# DEEP GRAPH SPECTRAL EVOLUTION NETWORKS FOR GRAPH TOPOLOGICAL TRANSFORMATION

## ABSTRACT

Characterizing the underlying mechanism of graph topological evolution from a source graph to a target graph has attracted fast increasing attention in the deep graph learning domain. However, there lacks expressive and efficient that can handle global and local evolution patterns between source and target graphs. On the other hand, graph topological evolution has been investigated in the graph signal processing domain historically, but it involves intensive labors to manually determine suitable prescribed spectral models and prohibitive difficulty to fit their potential combinations and compositions. To address these challenges, this paper proposes the deep Graph Spectral Evolution Network (GSEN) for modeling the graph topology evolution problem by the composition of newly-developed generalized graph kernels. GSEN can effectively fit a wide range of existing graph kernels and their combinations and compositions with the theoretical guarantee and experimental verification. GSEN has outstanding efficiency in terms of time complexity ($O(n)$) and parameter complexity ($O(1)$), where $n$ is the number of nodes of the graph. Extensive experiments on multiple synthetic and real-world datasets have demonstrated outstanding performance.

## 1 INTRODUCTION

Understanding the mechanism of graph generation and evolution has significant importance in many applications, such as brain simulation, mobility network simulation, and social network modeling and intervention. Beyond the traditional methods from network science domain, graph generation and evolution have been attracting fast increase attention by deep graph generative models due to their great potential of learning the underlying known generation and evolution mechanism in an end-to-end fashion Simonovsky & Komodakis (2018). Based on deep graph generative models, the graph generation problem is considered as decoding a graph based on latent variables following some underlying distribution while graph evolution can be modeled as a mapping to a target graph topology given a source graph topology. Graph evolution based on deep graph learning is a very challenging problem and is still in its nascent stage because of the extremely high-dimension of the data. An ideal model should be able to capture both the local and global characteristics of source graph and be able to determine the existence or weight of potential edges for each pair of nodes. Models that are both expressive and efficient are in urgent demand.

The domain which has investigated graph evolution for a long time is graph signal processing, where well-defined mathematical framework and various techniques such as graph wavelets and kernels that abstract the graph process in frequency domain have been hypothesized and verified in many applications. For example, Kunegis et al. have demonstrated that triangle-closing kernels fit very well to the evolution of graph spectrum during the "befriending process" in some social networks (Leskovec et al. (2008)). Most recently, neuroscience researchers found that the functional connectivity shares the same graph Fourier basis with structural connectivity in several special situations.

Although graph signal processing allows powerful and concise models to characterize many graph evolution processes, they require to first determine the potentially suitable type of graph kernel and then fit the parameters of it. However, this raises up serious challenges: First, it is difficult to discover or select suitable kernel types for various applications. Graph kernels are proposed based on the analyses and abstraction of the prior knowledge on various graph phenomena. But until now, quite a lot of phenomena have not yet been analyzed or interpreted by human. For example, it is unclear whether and how the spectrum of resting-state functional connectivity transforms into task-specific functional connectivity in human brain Hermundstad et al. (2013). Moreover, for many

sophisticated phenomena, the graph process typically involves the combination and composition of multiple graph processes corresponding to multiple kernels. For example, the evolution of social networks might involve not only the triangle closing process (i.e., two friends of a person tend to be friends) by triangle-closing kernels, but also could include the behavior diffusion process which can be characterized by diffusion kernels. Also, the involvement of different kernel process might be simultaneous or sequential, and hence prohibitively difficult to manually determine or combinatorially optimize.

To address these challenges, this paper proposes a novel end-to-end model named Deep Graph Spectral Evolution Network (GSEN) to optimally fit the graph evolution process by the composition of newly-developed generalized graph kernels. The generalized graph kernels widely cover existing graph kernels as well as their combination and composition as special cases, and hence are able to fit them with outstanding expressiveness. In addition to this high expressiveness, GSEN is also highly concise in terms of small parameter complexity and time complexity for training. Specifically, the number of parameters and memory complexity of GSEN are independent of the graph size while the time complexity for the training of GSEN is linear to the graph size. This largely outperforms the state-of-the-art, which typically requires $O(n^2)$ time complexity and memory complexity. Extensive experiments on several synthetic datasets and multiple real-world datasets in two domains have been conducted. The results demonstrate the superior accuracy of our GSEN over existing deep generative models for graph transformation and models based on graph signal processing. The higher efficiency of GSEN compared to existing deep generative models has also been verified.

## 2 RELATED WORK

### 2.1 SPECTRAL GRAPH TRANSLATION PROBLEMS

Spectral based approaches in graph translation have been the focus in many researches over the past decades. To model how networks are translated, the spectral evolution model was introduced by Kunegis et al. (2010). The growth of large networks is analyzed by studying the changes in the spectral characteristics of the graph. These changes are explained using the eigendecomposition of the graph adjacency matrix or its laplacian. The new link prediction approach shows how eigenvectors stay constant while the eigenvalues are evolved over the transition. This model can also generalize several graph kernels which are expressed as spectral transformations. Li et al. (2011) proposes the MERW (maximum entropy random walk) approach to the link prediction problem. MERW based approaches are introduced as various algorithms that could use four separate graph kernels, in addition to a class of similarity measures, to capture the proximity between two nodes. The resulting methods perform the prediction, while maintaining the centrality of the nodes. In Symeonidis et al. (2013), the link prediction problem for protein-protein interaction networks and online social networks is considered. The SpectralLink algorithm is proposed to compute the similarity between every two nodes, by exploiting the top few eigenvectors of the laplacian matrix, which eliminates the redundant and noisy information. The link prediction is then performed faster and more accurate. Variants of the aforementioned method are also derived for signed and directed graphs. Spectral Graph analysis has been useful in a behavior related link prediction problem Spiegel et al. (2011), where there's a need to predict whether and how much a user is likely to rate an item. Multiple network snapshots with temporal trends are captured and tensor factorization is used to extract hidden trends within a multi-dimension array. The higher-order data is then factorized into a lower dimension, using Parafac model. The spectral evolution model is finally applied, where the spectrum of decompositions change, while the eigenvectors stay constant.

### 2.2 DEEP LEARNING METHODS IN GRAPH SPECTRAL DOMAIN

There is a large body of research on deep graph learning, for tasks such as the embedding and classification of nodes and graphs. Kipf & Welling (2016) proposed a localized graph convolutional neural networks (CNNs) based on semi-supervised learning for graph-structured data, where labels are only available for a small subset of nodes. A neural network model is designed based on a layer-wise propagation rule. The model is then trained on the supervised target which includes all nodes with labels. A novel spectral graph CNN approach is proposed in Li et al. (2018) to graph data that varies in both size and connectivity. To capture the variation in the input graph topology, the training process includes applying a customized graph laplacian to each sample input. The laplacian then becomes trainable by parameterizing the distance metrics that measure vertex similarity. Deep convolutional approaches have been applied to data domains with irregularities which lack fundamental statistical properties in Henaff et al. (2015), to solve for large scale classification problems. In Def-

Table 1: Existing kernels for graph spectral translation problem

| Kernel Name | Matrix Function | Spectral Function |
|---|---|---|
| Laplacian Commute-time Kernel | $\mathcal{K}_{\mathrm{Com}}(L) = L^+$ | $U\Lambda^{-1}U^\intercal$, define $\Lambda_{i,i}^{-1} = 0$ if $\Lambda_{i,i} = 0$ |
| Normalized Laplacian Commute-time Kernel | $\mathcal{K}_{\mathrm{Com}}(Z) = Z^+$ | $U\Lambda^{-1}U^\intercal$, define $\Lambda_{i,i}^{-1} = 0$ if $\Lambda_{i,i} = 0$ |
| Normalized Adjacent Exponential Kernel | $\mathcal{K}_{\mathrm{Exp}}(N) = e^{\alpha N}$ | $Ue^{\alpha\Lambda}U^\intercal$ |
| Generalized Laplacian Kernel | $\mathcal{K}_{\mathrm{Gen}}(L) = (\sum_{k=0}^{\infty} \alpha_k L^k)^+$ | $U(\sum_{k=0}^{\infty} \alpha_k \Lambda^k)^{-1}U^\intercal$ |
| Generalized Normalized Laplacian Kernel | $\mathcal{K}_{\mathrm{Gen}}(Z) = \sum_{k=0}^{\infty} \alpha_k(I - Z)^k$ | $U(\sum_{k=0}^{\infty} \alpha_k(I - \Lambda)^k)U^\intercal$ |
| Heat Diffusion Kernel | $\mathcal{K}_{\mathrm{Heat}}(L) = e^{-\alpha L}$ | $Ue^{-\alpha\Lambda}U^\intercal$ |
| Normalized Heat Diffusion Kernel | $\mathcal{K}_{\mathrm{Heat}}(Z) = e^{-\alpha Z}$ | $Ue^{-\alpha\Lambda}U^\intercal$ |
| Normalized Adjacent Neumann Kernel | $\mathcal{K}_{\mathrm{Neu}}(N) = (I - \alpha N)^{-1}$ | $U(I - \alpha N)^{-1}U^\intercal$ |
| Normalized Adjacent Path Count Kernel | $\mathcal{K}_{\mathrm{Path}}(N) = \sum_{k=0}^{\infty} \alpha_k N^k$ | $\sum_{k=0}^{\infty} \alpha_k(UD^{-\frac{1}{2}}\Lambda D^{-\frac{1}{2}}U^\intercal)^k$ |
| Regularized Laplacian Kernel | $\mathcal{K}_{\mathrm{Reg}}(N)(I + \alpha N)^{-1}$ | $U(I + \alpha\Lambda)^{-1}U^\intercal$ |
| Normalized Regularized Laplacian Kernel | $\mathcal{K}_{\mathrm{Reg}}(Z)(I + \alpha Z)^{-1}$ | $U(I + \alpha\Lambda)^{-1}U^\intercal$ |

ferrard et al. (2016), CNNs are presented in the context of spectral graph theory, and fast localized convolutional filters are designed.

## 2.3 DEEP LEARNING METHODS FOR GRAPH TRANSFORMATION PROBLEMS

Graph Transformation modeling based on deep neural networks has attracted fast-increasing attention recently, where existing methods are based on spatial domain by operating the explicit connectivity among the nodes (Guo et al. (2019), Guo et al. (2018), Do et al. (2019)). The prediction in most cases is performed either on the node attributes of the graph or its topology while the other is fixed. Guo et al. (2019) proposes the NEC-DGT (Node-Edge Co-evolving Deep Graph Translator) framework as a novel technique to approach the simultaneous prediction challenge. A portion of these research is only tailored for specific applications and domains (Do et al. (2019)). For example, Do et al. (2019) and Jin et al. (2018) proposed methods only for transferring molecule graphs. Spatio-temporal dependencies in traffic flow are modeled as a diffusion process in a directed graph through a DCRNN (Diffusion Convolutional Recurrent Neural Network) model Li et al. (2017). Using bidirectional random walks and encoder-decoder architecture the spatial and temporal dependencies are captured respectively. However, until now there is no work in this domain that models the graph topological transformation in spectral domain.

## 3 GRAPH TOPOLOGY TRANSFORMATION VIA SPECTRAL EVOLUTION

This paper focuses on a problem of predicting the topology of a target graph based on that of a source graph by characterizing spectral graph evolution.

### 3.1 PROBLEM FORMULATION

Define a source graph as an undirected weighted graph $G = (V, E, A)$ where $V$ is the set of nodes with size of $|V|$, $E \subseteq V \times V$ is the set of edges, and $A \in \mathbb{R}^{|V| \times |V|}$ is the adjacent matrix that defines the weights of the edges. The adjacent matrix $A$ can be normalized by defining the normalized adjacent matrix $N = D^{-\frac{1}{2}}AD^{-\frac{1}{2}}$, where $D \in \mathbb{R}^{|V| \times |V|}$ is the diagonal matrix where each diagonal element is the degree of corresponding node. Besides, the Laplacian matrix of the source graph $G$ is defined as $L = D - A$, and the normalized Laplacian matrix is defined as $Z = D^{-\frac{1}{2}}LD^{-\frac{1}{2}} = I - N$ where $I$ is the identity matrix. Define graph spectrum as $\Lambda \in \mathbb{R}^{|V| \times |V|}$ and graph Fourier basis as $U$, which are obtained from the eigendecomposition of Laplacian matrix $L = U\Lambda U^\intercal$. A target graph is defined as $G' = (V', E', A')$, where the set of nodes $V' = V$, edges $E'$, adjacent matrix $A'$, normalized adjacent matrix $N'$, the Laplacian matrix $L'$, the normalized Laplacian matrix $Z'$, graph spectrum $\Lambda'$ and Fourier basis $U'$ are defined the same way as that of source graph.

**Graph Transformation via Spectral Evolution:** The spectral graph translation problem states that the graph topological transformation $G' \leftarrow F(G)$ from a source graph $G$ to target graph $G'$ can be modeled by a change in the graph's spectrum, while the graph basis remains the same.

To determine the function $F$, various graph kernels based on the existing research on graph wavelets can be utilized, including heat kernels $\mathcal{K}_{\mathrm{HEAT}}(L) = \exp(-\alpha L)$ and many others such as those listed in Table 1. Several such graph kernels have been empirically demonstrated to model some specific graph process effectively. For example, Kunegis et al. has verified that path count kernels fit very well to the link prediction problem in some email networks (Kunegis et al. (2010)). The evolutions of social networks typically involve triangle closing process (i.e., two friends of a person tend to be friends), which has been verified to be effectively modeled by triangle-closing kernels Leskovec et al. (2008).

Existing techniques basically first determine the potentially suitable type of kernels and then fit the parameters of the kernels. However, this raises up several challenges: First, it is difficult to discover or select suitable kernel types for various applications. Graph kernels are proposed based on the

analyses and abstraction of the prior knowledge on various graph phenomena. But until now, quite a lot of phenomena have not yet been analyzed or interpreted by human, let alone formulated kernel functions. For example, it is unclear whether and how the spectrum of resting-state functional connectivity transforms into task-specific functional connectivity in human brain Hermundstad et al. (2013). Moreover, for many sophisticated phenomena, the graph process typically involves additive and/or sequential compositions of multiple graph processes described by multiple kernels. For example, the evolution of social network might involve not only triangle closing process (i.e., two friends of a person tend to be friends) by triangle-closing kernels, but also could include behavior diffusion process which can be characterized by diffusion kernels. Also, the involvement of different kernel process might be simultaneous or sequential, and hence prohibitively difficult to manually determine or combinatorially optimize.

## 3.2 GENERALIZED GRAPH KERNELS

In order to address the above mentioned challenges, we propose a new generalized graph kernel that is highly expressive to cover various graph kernels as well as their compositions. We first formulate the learning of such expressive kernel as an optimization problem as follows.

**Lemma 3.1.** *Without loss of generalizability, using graph Laplacian $L$ and $L'$ to represent the graph topology of $G$ and $G'$, the spectral graph translation can be explicitly formulated as $F(L) \to L'$ by an analytic function $F$, which is learned by the following equation given $[F(\Lambda)]_{kk} = f(\Lambda_{kk})$:*

$$\min_f \sum_k (f(\Lambda_{k,k}) - U_{\cdot,k}^{\mathsf{T}} L' U_{\cdot,k})^2 \tag{1}$$

*Proof.* The training purpose of $F(\cdot)$ is to minimize the squared loss against the real target graph:

$$\min_F \|F(L) - L'\|_2^2 = \|F(U\Lambda U^{\mathsf{T}}) - L'\|_2^2$$

$$= \|\sum_{k=0}^{\infty} \frac{F^{(k)}(\gamma)}{k!}(U\Lambda U^{\mathsf{T}})^k - L'\|_2^2 \quad \text{(power expansion)}$$

$$= \|U\left(\sum_{k=0}^{\infty} \frac{F^{(k)}(\gamma)}{k!}(\Lambda)^k\right)U^{\mathsf{T}} - L'\|_2^2$$

$$= \|UF(\Lambda)U^{\mathsf{T}} - L'\|_2^2$$

$$= \|U \cdot diag([F(\Lambda_{1,1}), \cdots, F(\Lambda_{|V|,|V|})])U^{\mathsf{T}} - L'\|_2^2$$

$$= \|diag([F(\Lambda_{1,1}), \cdots, F(\Lambda_{|V|,|V|})]) - U^{\mathsf{T}} L' U\|_2^2$$

Hence, this problem is equivalent to the problem of $\min_f \sum_k (f(\Lambda_{k,k}) - U_{\cdot,k}^{\mathsf{T}} L' U_{\cdot,k})^2$ given $F(\Lambda)_{kk} = f(\Lambda_{kk})$. The proof is completed. $\square$

We propose the following new generalized graph kernel:

$$F(\Lambda) = \sum_{k=1}^{\infty} \alpha_k \Lambda^k + \gamma_k D^{-k} \Lambda^k + \beta I \tag{2}$$

Here we introduce some important properties of the proposed generalized graph kernel.

**Lemma 3.2.** *The generalized graph kernel in Equation 2 has the following properties:*

1. *Existing graph kernels based on $L, A, N$, and $Z$ such as those in Table 1 are its special cases.*

2. *The additive combinations and compositions of the existing graph kernels are special cases of our operation.*

*Proof.* Now we prove Property 1. Graph kernels are typically functions of four types of variables, namely Laplacian $L$, adjacency matrix $A = D - L$, normalized Laplacian $Z = D^{-1/2} L D^{-1/2}$, and normalized adjacency matrix $N = I - D^{-1/2} L D^{-1/2}$. For those kernels based on $L$ and $A$, they can be transformed to $[F(\Lambda)]_{i,i} = \sum_{k=0}^{\infty} \frac{m_{k,i}}{k!} \Lambda_{i,i}^k$, where $m_{k,i} = f^{(k)}(0)$ for Laplacian while $m_{k,i} = f^{(k)}(D_{i,i})$ for adjacency matrix. Therefore, both of them can be fit by our generalized graph kernel by setting $\gamma_k := 0$ for all $k = 0, 1, \cdots$. For those kernels based on $N$ and $Z$, they can be transformed to $[F(\Lambda)]_{i,i} = \sum_{k=0}^{\infty} \frac{m}{k!}(D_{i,i}^{-k} \Lambda_{i,i})^k$, where $m = 0$ for $Z$ while $m = 1$ for $N$. Hence, both of them can be fit by setting $\alpha_k := 0$ for all $k = 0, 1, \cdots$.

Here we prove Property 2. Assume there are two generalized graph kernels $F_a(\Lambda)$ and $F_b(\Lambda)$, then it is easy to see that both their sumation and composition can still be fit by another generalized graph

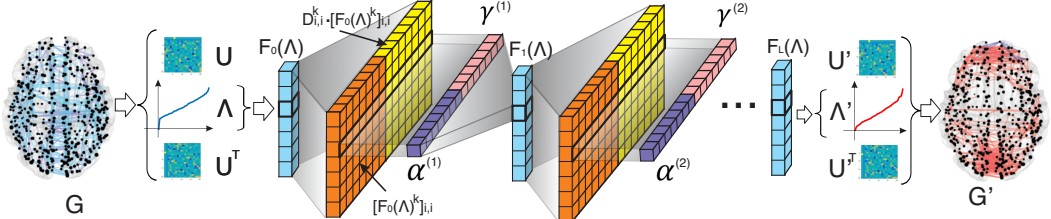

Figure 1: The architecture of Deep Graph Spectral Evolution Networks.

kernel. By leveraging Property 1, the additive and composition of all the various graph kernels such as those in Table 1 can be fit by our generalized graph kernels. □

## 4 DEEP GRAPH SPECTRAL EVOLUTION NETWORKS

In this section, a new neural network named GSEN, which is based on the proposed generalized graph kernel, is proposed. To achieve this, we reduce the order of the polynomials from infinity to $K$ that is independent of and typically far less than the graph size. Moreover, our GSEN is composed by stacking multiple such generalized graph kernel as a special type of multi-order 1-D convolution operation, as illustrated in Figure 1 and described as follows.

Specifically, each layer can be expressed as follows:

$$F_l(\Lambda) = H_l \left( \sum_{k=1}^{\infty} (\alpha_k I + \gamma_k D^{-k}) F_{l-1}(\Lambda)^k + \beta I \right) \quad (3)$$

where the function $H_l(\cdot)$ is an activation function which performs element-wise activation based on commonly used ones such as ReLU, sigmoid, or linear. An equivalent scalar form of the above equation is expressed as $f_l(\Lambda_{i,i}) = h_l \left( \sum_{k=1}^{\infty} (\alpha_k + \gamma_k D_{i,i}^{-k}) f_{l-1}(\Lambda_{i,i})^k + \beta \right)$, where $h_l(\cdot)$ is a scalar version of $H_l(\cdot)$.

As shown in Figure 1, we implement GSEN through an $L$-layer convolution operations from the source graph to target graph. Specifically, the input, namely $F_0(\Lambda)$, is $\Lambda$ that is the spectrum of the source graph. For the $l-1$th layer, the diagonal vectors of the matrices $I, F_{l-1}(\Lambda), F_{l-1}(\Lambda)^2, \cdots, F_{l-1}(\Lambda)^K$ are calculated and concatenated as shown in the orange region in Figure 1. Similarly, the diagonal vectors of the matrices $I, D \cdot F_{l-1}(\Lambda), D^2 \cdot F_{l-1}(\Lambda)^2, \cdots, D^K \cdot F_{l-1}(\Lambda)^K$ are calculated and concatenated as shown in the yellow region in Figure 1. Then these two regions are convoluted by the kernels $\alpha^{(l)}$ and $\gamma^{(l)}$, respectively, to obtain $F_l(\Lambda)$ after performing activation function. Such convolution operation is repeatedly performed until $L$-th layer, which outputs the predicted graph spectrum $F_L(\Lambda)$ for the target graph.

**Complexity and Efficiency:** Training neural network amounts to solve the optimization problem in Equation 1, which can be handled by backpropagation. Our method largely and effectively reduces the number of parameters, to $2 \cdot K \cdot L$, which is small and independent of the size of the graph and hence is highly memory-efficient and scalable. In terms of the time complexity, the calculation of the powers of graph spectrum has a time complexity of $O(K \cdot N \cdot L)$ while the convolution operations involves another $O(K \cdot N \cdot L)$ so the total time complexity of the neural network is $O(K \cdot N \cdot L)$. Also, notice the generation of the input data involves eigendecomposition of $L$ which could be time-consuming for large graph. To address this issue, we can leverage reduced eigendecomposition to only involve the calculation of lower-rank matrix and hence can largely speed up and scalable to large graph.

## 5 EXPERIMENT

In this section, the experimental settings are first introduced, then the performance of the proposed method is presented through a set of comprehensive experiments. All the experiments are conducted on a 64-bit machine with 40 GB memory, a 4-core Intel ® CPU and an Nvidia ® RTX-2080 Ti GPU. The proposed method is implemented with Pytorch deep learning framework.

### 5.1 EXPERIMENTAL SETUP

We evaluate the effectiveness on 11 synthetic datasets, and 4 real-world datasets on brain network prediction and malware confinement in the Internet of Things (IoT) task. The datasets, evaluation methods, and comparison methods are elaborated in turn.

#### 5.1.1 DATASETS

• **Synthetic Datasets:** In each of the 11 synthetic datasets, we generate 1000 source-target graph pairs. Specifically, first, 1000 unweighted and undirected random graphs with 50 nodes and 200 edges are generated as the source graphs, then each edge in the source graphs is assigned with a

random weight between 0 and 1. Finally, 1000 target graphs are generated by applying one of the kernels depicted in Table 1.

• **Real-world HCP Dataset:** In these datasets, the source and the target graphs respectively reflect the structural connectivity (SC) and the functional connectivity (FC) of the same subject's brain network. In particular, both types of connectivity are processed from the Magnetic Resonance Imaging (MRI) data obtained from the human connectome project (HCP) Van Essen et al. (2013). By following the preprocessing procedure in Wang et al. (2019), the SC data is constructed by applying probabilistic tracking on the diffusion MRI data using the Probtrackx tool from FMRIB Software Library Jenkinson et al. (2012) with 68 predefined regions of interests (ROIs). Then, the FC is defined as the Pearson's correlation between two ROIs' blood oxygen level-dependent time obtained from the resting-state functional MRI data. Finally, all the 823 pairs of SC and FC adjacent matrices are normalized as defined in Section 3.

• **Real-world IoT Datasets:** In these datasets, the nodes represent the Internet of Things (IoT) devices and the edges denote the communication links between two devices. Each source graph reflects the communication status of the network, and some of the nodes in the network are infected by some types of malware. To limit the devices that are infected by the malware propagating to other devices, the malware confinement is conducted by cutting some of the links while maximizing the functionality of the network. The confined network is considered as the target graph that corresponds to the source graph. The IoT datasets contain three datasets, namely IoT-20, IoT-40, and IoT-60, which include 20, 40, and 60 devices. There are 343 source-target graph pairs in each of IoT datasets.

### 5.1.2 COMPARISON METHODS

The comparison methods include: **Graph spectral transformation kernels:** We compared our GSEN method with all single kernel methods defined in Table 1 on synthetic datasets. The parameters $\alpha$ or $\{\alpha_k\}_{k=1}^K$ in Table 1 is learned from the training data. **Baseline method:** For this method, the eigenvalue transformation function $\mathcal{F} : \Lambda \to \Lambda'$ is learned by a fully connected four-layer perceptron activated by tanh function. Each hidden layer contains $4n$ neurons, where $n$ is the number of nodes in the graph. **Brain network prediction methods:** We consider four classic brain network prediction methods that use SC to FC Galán (2008); Abdelnour et al. (2014); Meier et al. (2016); Abdelnour et al. (2018). Abdelnour et al. (2014) and Abdelnour et al. (2018) considered the graph spectral transformation kernels by assuming that SC and FC share the identical eigenvectors on their Laplacians. The remaining two methods directly consider the graph translation between SC and FC. **GT-GAN**: Graph Translation-Generative Adversarial Networks (GT-GAN) by Guo et al. (2018) is a newly proposed general-purpose graph topology translation method based on the graph generative adversarial network. **C-DGT** : node-edge Co-evolving Deep Graph Translator (C-DGT) by Guo et al. (2019) is the state-of-the-art deep graph translation network, which considers both and edge attributes that are regularized in the spectral domain. For the datasets without node and edge attributes in our experiments, we simply assign the attributes as all-ones.

### 5.1.3 EVALUATION METRIC

For the effectiveness experiments, the Pearson correlation is computed between the upper triangular values of the normalized adjacent matrix of the real target graph and that of the predicted target graph. For all the comparison and our methods, 5-fold cross-validation is performed.

For the efficiency experiments, as the training time depends on the data and the max number of epochs for the gradient-based optimization algorithms (e.g. SGD, ADAM). We use the per-epoch training time on CPU as the evaluation metric for efficiency study.

### 5.2 PERFORMANCE

In this section, the performance of the proposed method, namely GSEN, as well as other methods on effectiveness and efficiency on both 11 synthetic and 4 real-world datasets are elaborated. In addition, the case studies and the sensitivity tests on the real-world datasets are also presented.

### 5.2.1 PERFORMANCE ON SYNTHETIC DATASETS

For synthetic datasets, we compare the Pearson correlation between the target graph generated by various kernels and the graphs predicted by various methods. Table 2 summarizes the effectiveness comparison for 11 synthetic datasets. Our GSEN method achieves 0.90 Pearson correlation on average among all 11 synthetic datasets. In contrast, the second-best method, namely the C-DGT, only can achieve 0.61 Pearson correlation. The traditional graph spectral kernel functions can achieve

| Method | Dataset | | | | | | | | | | | |
|---|---|---|---|---|---|---|---|---|---|---|---|---|
| | $\mathcal{K}_{\text{Com}}(L)$ | $\mathcal{K}_{\text{Com}}(Z)$ | $\mathcal{K}_{\text{Exp}}(N)$ | $\mathcal{K}_{\text{Gen}}(L)$ | $\mathcal{K}_{\text{Gen}}(Z)$ | $\mathcal{K}_{\text{Heat}}(L)$ | $\mathcal{K}_{\text{Heat}}(Z)$ | $\mathcal{K}_{\text{Neu}}(N)$ | $\mathcal{K}_{\text{Path}}(N)$ | $\mathcal{K}_{\text{Reg}}(L)$ | $\mathcal{K}_{\text{Reg}}(Z)$ | Avg. |
| $\mathcal{K}_{\text{Com}}(L)$ | **1.00** (GS) | **1.00** (GS) | 0.28 | -0.04 | -0.15 | -0.26 | -0.26 | 0.23 | 0.26 | -0.02 | -0.23 | 0.16 |
| $\mathcal{K}_{\text{Com}}(Z)$ | **1.00** (GS) | **1.00** (GS) | 0.28 | -0.04 | -0.15 | -0.26 | -0.26 | 0.23 | 0.26 | -0.02 | -0.23 | 0.16 |
| $\mathcal{K}_{\text{Exp}}(N)$ | 0.23 | 0.23 | **1.00** (GS) | 0.25 | 0.69 | **1.00** | **1.00** | -0.88 | -0.91 | 0.13 | -0.88 | 0.17 |
| $\mathcal{K}_{\text{Gen}}(L)$ | -0.11 | -0.28 | **-1.00** | **0.93** (GS) | 0.83 | 0.06 | 0.99 | -0.94 | -0.99 | 0.80 | 0.95 | 0.11 |
| $\mathcal{K}_{\text{Gen}}(Z)$ | -0.02 | -0.18 | -0.96 | -0.05 | **1.00** (GS) | 0.21 | 0.84 | -0.84 | -0.92 | -0.02 | 0.98 | 0.00 |
| $\mathcal{K}_{\text{Heat}}(L)$ | 0.23 | 0.23 | **1.00** | 0.25 | 0.69 | **1.00** (GS) | **1.00** (GS) | -0.88 | -0.91 | 0.13 | -0.88 | 0.17 |
| $\mathcal{K}_{\text{Heat}}(Z)$ | 0.23 | 0.23 | **1.00** | 0.25 | 0.69 | **1.00** (GS) | **1.00** (GS) | -0.88 | -0.91 | 0.13 | -0.88 | 0.17 |
| $\mathcal{K}_{\text{Neu}}(N)$ | 0.25 | 0.29 | 0.93 | -0.08 | -0.76 | -0.87 | -0.99 | **1.00** (GS) | 0.98 | 0.02 | -0.91 | -0.01 |
| $\mathcal{K}_{\text{Path}}(N)$ | 0.01 | 0.29 | 0.95 | 0.03 | -0.77 | -0.18 | -0.99 | **1.00** | **1.00** (GS) | 0.01 | -0.91 | 0.04 |
| $\mathcal{K}_{\text{Reg}}(L)$ | -0.02 | -0.23 | -0.99 | 0.36 | 0.96 | -0.11 | 0.91 | -0.91 | -0.97 | **0.97** (GS) | **1.00** | 0.09 |
| $\mathcal{K}_{\text{Reg}}(Z)$ | -0.02 | -0.23 | -0.99 | 0.31 | 0.97 | -0.10 | 0.91 | -0.91 | -0.97 | 0.65 | **1.00** (GS) | 0.06 |
| GT-GAN | 0.12 | 0.18 | 0.26 | 0.00 | 0.93 | 0.48 | 0.25 | 0.53 | 0.69 | 0.18 | 0.53 | 0.38 |
| C-DGT | -0.05 | 0.16 | **1.00** | -0.02 | 0.80 | **1.00** | **1.00** | 0.92 | 0.98 | -0.02 | 0.92 | 0.61 |
| Baseline | -0.14 | -0.28 | -0.99 | -0.05 | 0.27 | 0.07 | 0.17 | 0.76 | 0.99 | -0.03 | 0.63 | 0.13 |
| GSEN | 0.97 | 0.72 | 0.99 | 0.80 | **1.00** | 0.89 | 0.99 | **1.00** | **1.00** | 0.71 | 0.85 | **0.90** |

Table 2: Pearson correlation between the predicted graph and empirical graph 11 synthetic datasets. Each column denotes the Pearson correlation of the synthetic dataset generated by the kernel function of the second row. Each row denotes the Pearson correlation of the prediction method of the first column. Some results are "gold standard" ones because the predictor and synthetic data generator use the same graph kernel, and hence are marked as "GS" for those results. The right-most column denotes the average Pearson correlation among all 11 synthetic datasets. The highest Pearson correlation in each column/dataset is highlighted in bold font while the second-highest Pearson correlation is marked with an underline.

near-perfect results on the synthetic datasets generated from the same kernel, so all the diagonal Pearson correlation coefficients from the top left corner in Table 2 are close to 1. But none of these kernels perform well on all 11 synthetic datasets, which means they are not generic methods. Their average performance is thus worse than the deep learning-based methods. The deep learning-based graph translation method C-DGT performs much better than the other deep learning-based GT-GAN and fully-connected baseline methods. This because the C-DGT methods partially consider the spectral property as a regularization term such that it can have relatively good performance (e.g. $> 0.7$) on 7 out of 11 synthetic datasets, but not as good as our generic spectral method. GSEN typically performs better on normalized Laplacian matrix $Z$ than original Laplacian matrix $L$, because the eigenvalues of the normalized Laplacian matrix are between 0 and 2, which can have a good estimation when using Taylor expansion to estimate $\mathcal{F}(\Lambda)$.

### 5.2.2 PERFORMANCE ON REAL-WORLD DATASETS

• **Metric-based evaluation:** Table 3 shows the Pearson coefficient by comparing the predicted graphs with the empirical target graphs. Our method achieves the highest Pearson coefficient on 3 out of 4 datasets, and the highest average Pearson correlation among the four real-world datasets. For the malware confinement datasets, namely the IoT datasets, GSEN slightly outperforms the C-DGT method, which is the state-of-the-art method on these datasets. But as we will show in the next section, our method is at least 40 times faster than the C-DGT method. In addition, the C-DGT method receives the lowest Pearson correlation coefficient on the brain network SC-FC translation dataset whose nodes attributes are not available.

| Method | Dataset | | | | |
|---|---|---|---|---|---|
| | IoT-20 | IoT-40 | IoT-60 | SC-FC | Avg. |
| Galan2008 | 0.74 | 0.79 | 0.81 | 0.23 | 0.64 |
| Abdelnour2014 | 0.73 | 0.76 | 0.81 | 0.23 | 0.63 |
| Meier2016 | 0.74 | 0.78 | 0.81 | 0.26 | 0.65 |
| Abdelnour2018 | 0.73 | 0.76 | 0.81 | 0.23 | 0.63 |
| GT-GAN | 0.80 | 0.74 | 0.64 | **0.45** | 0.66 |
| C-DGT | 0.81 | 0.82 | **0.84** | 0.14 | 0.65 |
| Baseline | 0.70 | 0.72 | 0.74 | 0.33 | 0.62 |
| GSEN | **0.82** | **0.84** | **0.84** | 0.35 | **0.71** |

Table 3: Pearson correlation between the predicted graph and empirical graph on real-world datasets

The GT-GAN method achieves the highest SC-FC dataset, but performs worse than C-DGT and GSEN on other three datasets. The GT-GAN method is also the slowest method in terms of the per epoch training time. None of the methods can achieve the Pearson correlation higher than 0.5 including the top four methods that are exclusively designed for the SC-FC mapping problem in the neural science domain. This might be caused by the noise in the resting-state fMRI data. We will show some insightful reasons through multiple case studies below.

• **Case study on the brain network SC-FC prediction dataset:**

Figure 2 plots two subject's 1) structural connectivity (i.e., the adjacent matrix of the source graph shown on the left column), 2) empirical functional connectivity (i.e., the adjacent matrix of target graph shown on the middle column), 3) predicted functional connectivity (i.e., the adjacent matrix of target graph shown on the right column.

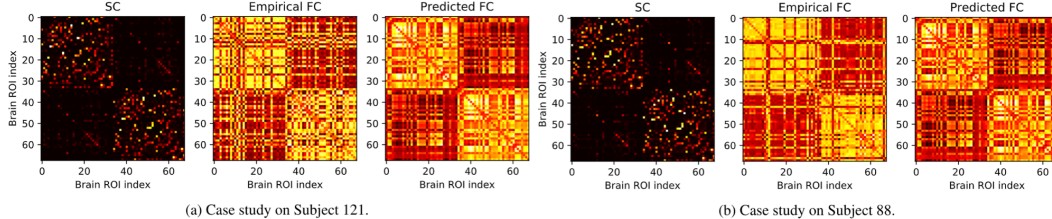

(a) Case study on Subject 121.        (b) Case study on Subject 88.

Figure 2: Case studies of the source, real, and predicted target graph topologies on SC-FC dataset.

As shown in Figure 2, the predicted FC using Subject 121's SC is very close to the same subject's empirical FC. On the other hand, the predicted FC using Subject 88's SC is different from Subject 88's empirical FC, although Subject 88's SC is very similar to Subject 121's SC. This is because SC reflects human brain's anatomical neural network, which has relatively less individual differences among human beings. Unlike SC, the FC used in this datasets reflects the Pearson correlations between two time series (i.e., Blood Oxygen Level Dependent (BOLD) signal) of different brain Regions Of Interests (ROIs), when the subject is instructed under the resting-state. In practice, it is difficult to control these subjects' brain activities, which causes the empirical FC very noisy such that may affect the performance of all prediction methods. The additional cases are provided in our supplementary material due to the space limitation.

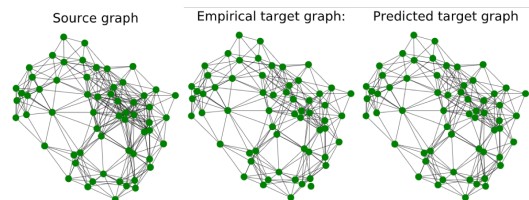

Figure 3: Case studies of the source, real, and predicted target graph topologies on IoT dataset.

• **Case study on the malware confinement (IoT) dataset:** Figure 3 demonstrates the cases of the source graphs, empirical target graph and the predicted target graphs by our method from all three malware confinement datasets. The leftmost source graph in each dataset denotes the original network connections. To prevent the network ceased by malware, some of the links in the network are cut while maintaining the optimal functionality of the entire network, which formulates the empirical target graph that is sparser than the source graph. The rightmost graph is the target network predicted by GSEN using the source graph. When comparing the empirical target graph with the predicted target graph, it is obvious that our method can mostly predict which link should be cut to prevent malware propagation.

### 5.2.3 EFFICIENCY EVALUATION

To validate the efficiency as well as the scalability of the proposed method, we use three real-world IoT datasets whose number of nodes is from 20 to 60. We further enlarge the IoT-20 dataset from 200 to 1000 nodes, which generates four larger datasets, namely the IoT-200, · · ·, IoT-1000 datasets. We report the results in Table 4 for the mean training time per epoch using CPU for 100 epochs on the aforementioned 7 datasets. We compare the results with the two deep learning-based graph translation methods. For our network, we set both the degree of power $K$ and the number of layers to 5. For the other two comparison methods, the default

| Dataset | GSEN time | GSEN speed up | GT-GAN time | GT-GAN speed up | C-DGT time | C-DGT speed up |
|---|---|---|---|---|---|---|
| IoT-20 | 0.06s | × 1 | 31s | × 517 | 2.44s | × 41 |
| IoT-40 | 0.09s | × 1 | 66s | × 733 | 5.86s | × 65 |
| IoT-60 | 0.13s | × 1 | 108s | × 831 | 12.10s | × 93 |
| IoT-200 | 0.21s | × 1 | 174s | × 829 | 40s | × 190 |
| IoT-400 | 0.72s | × 1 | 692s | × 961 | - | - |
| IoT-600 | 1.60s | × 1 | 1611s | ×1007 | - | - |
| IoT-800 | 2.89s | × 1 | 2964s | ×1026 | - | - |
| IoT-1000 | 4.75s | × 1 | 4112s | ×866 | - | - |

Table 4: Training time per epoch. (-) indicates out-of-memory error.

settings are applied. As shown in Table 4, our method is on average 967 times faster than the GT-GAN method and 72 times faster than the C-DGT method. Notice that the C-DGT is unable to handle the graphs with more than 400 nodes due to the out-of-memory error. The scalability of our GSEN is remarkable, which can be trained in 4.75 seconds per epoch on the graphs with 1000 nodes.

## 6 CONCLUSIONS

This paper focuses on the problem of spectral graph topological evolution, by proposing a novel deep Graph Spectral Evolution Networks (GSEN) which achieves a compelling trade-off between model expressiveness and efficiency. The proposed GSEN solves crucial drawbacks of the existing models in the graph topological evolution domain, which typically suffer from superlinear time and memory complexity. Experimental results on multiple synthetic and real-world datasets demonstrated the outstanding expressiveness and efficiency accuracy in terms of the graph topology prediction accuracy and runtime, as well as qualitative analyses on the predicted graph topologies.

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

## A    SUPPLEMENTARY MATERIAL

This is the supplementary material for deep graph spectral evolution networks for graph topological evolution. In this supplementary material we provide additional case study results on the brain network structural connectivity (SC) and functional connectivity (FC) datasets.

### A.1    HYPER-PARAMETER SENSITIVITY TEST

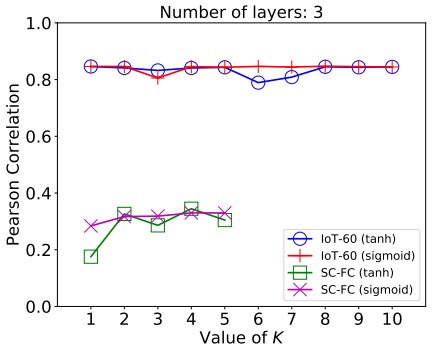
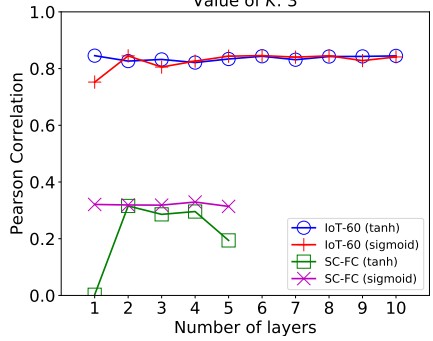

(a) Sensitivity test to the power degree $K$      (b) Sensitivity test to the number of layers

Figure 4: Sensitivity analysis

We use two real-world datasets, namely the IoT-60 dataset and the SC-FC dataset to test the sensitivity of the proposed method to its hyper-parameters. The proposed method includes three hyper-parameters: 1) the number of layers, 2) the degree of power $K$, and 3). the activation function. For the choice of activation function, we exclusively select the sigmoid($\cdot$) and tanh($\cdot$) function because they can easily constrain the eigenvalues to a valid range (e.g. [0,2) for the normalized Laplacian matrix). The results of the sensitivity test to the power degree $K$ are plotted in Figure 4a, which fix the number of layers to 3. Our method is quite robust for $K > 1$ on both datasets using both activation functions. When $K = 1$, the Pearson correlations are dropped on the SC-FC datasets because of the inaccurate estimation of the Taylor expansion. On the other hand, the sensitivity tests for the number of layers are plotted in Figure 4b. The propose method is also insensitive to the number of layers as long as the number of layers is greater than one. Recall when there is only one layer, our method is reduced to a single graph spectral kernel that may not be able to handle sophisticated graph translation problem (e.g. the SC-FC prediction problem). Therefore, this result has validated the necessity of the "deep" proposed in this paper.

### A.2    ADDITIONAL CASE STUDY RESULTS ON SC-FC DATASETS

The additional case study results for SC-FC datasets are shown in Figure 5, and the case study results for IoT datasets are shown in Figure 6.

### A.3    ADDITIONAL PERFORMANCE ON REAL-WORLD DATASETS

Table 5 shows the R2 coefficient by comparing the predicted graphs with the empirical target graphs. R2 is a metric positively related to the proportion of the variance in the dependent variable that is predictable from the independent variable(s). Thus, the higher R2 is, the better the performance will be. Similar to the results in Pearson correlation in Table 3, our method achieves the highest R2 coefficient on 3 out of 4 datasets, and the highest average R2 correlation among the four real-world datasets. For the malware confinement datasets, namely the IoT datasets, our method outperforms the C-DGT method in two out of three datasets. Also note that our method is at least 40 times faster than the C-DGT method. Our method also achieves highest R2 score on brain network dataset (i.e., the column of "SC-FC" in Table 5. In all, our method GSEN achieves the highest performance in general among all four datasets with large margin comparing to the second-best one which is the baseline method.

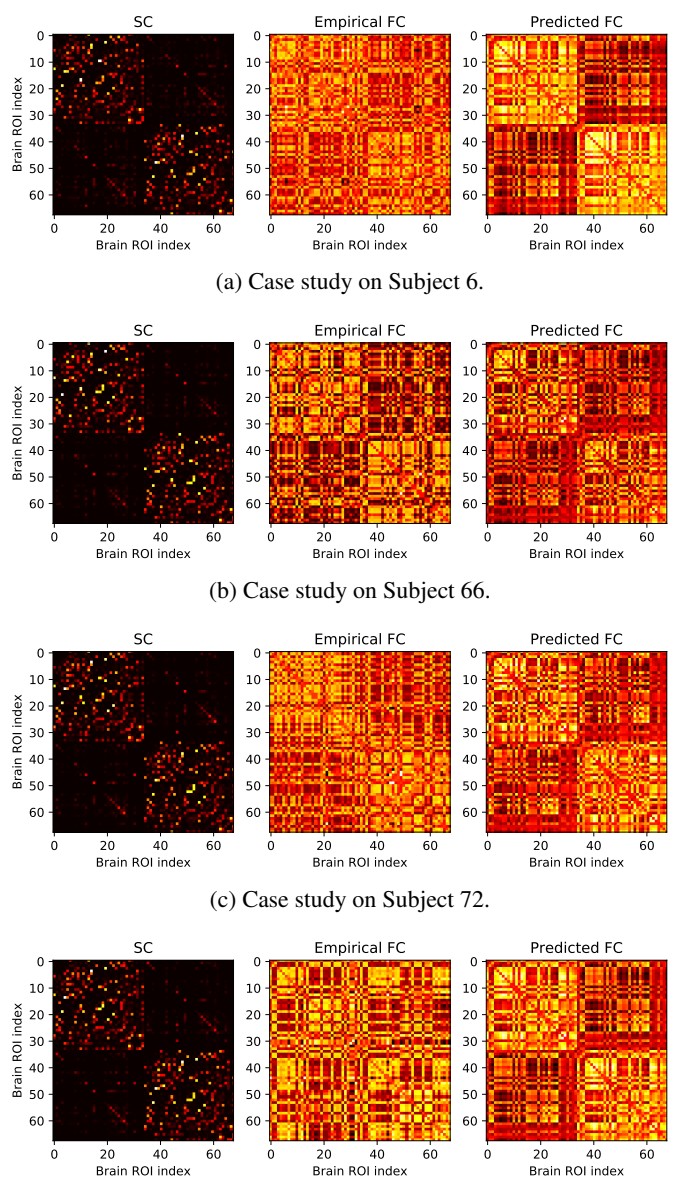

(a) Case study on Subject 6.

(b) Case study on Subject 66.

(c) Case study on Subject 72.

(d) Case study on Subject 123.

Figure 5: Case study of brain functional connectivity predicted by our method.

| Method | Dataset | | | | |
| | IoT-20 | IoT-40 | IoT-60 | SC-FC | Avg. |
|---|---|---|---|---|---|
| Galan2008 | 0.5433 | 0.6044 | 0.6527 | -5.7832 | -0.9957 |
| Abdelnour2014 | -1.8336 | -0.0002 | -0.6364 | -0.8801 | -0.8376 |
| Meier2016 | 0.5386 | 0.5991 | 0.6525 | -3.5465 | -0.4391 |
| Abdelnour2018 | -0.0011 | -0.0005 | -0.0002 | -0.8805 | -0.2206 |
| GT-GAN | **0.6552** | 0.4809 | 0.1795 | -1.0315 | 0.0710 |
| C-DGT | 0.6400 | 0.6716 | 0.7050 | -4.1410 | -0.5311 |
| Baseline | 0.4051 | 0.4601 | 0.5137 | -0.7548 | 0.1560 |
| GSEN | 0.6275 | **0.7050** | **0.7062** | **-0.5847** | **0.3635** |

Table 5: R2 score between the predicted graph and empirical graph on real-world datasets

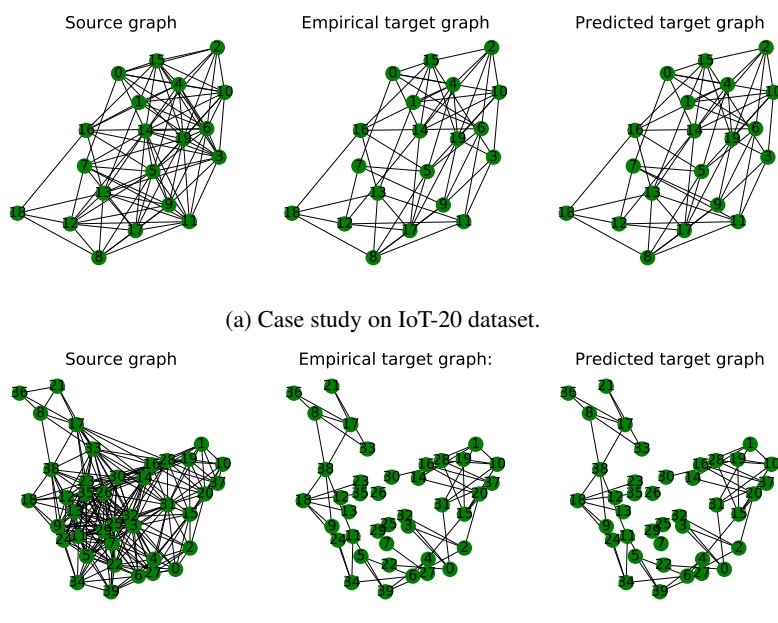

(a) Case study on IoT-20 dataset.

(b) Case study on IoT-40 dataset.

Figure 6: Case study of malware confinement datasets with our method.

