# OpenReview forum: "DEEP GRAPH SPECTRAL EVOLUTION NETWORKS FOR GRAPH TOPOLOGICAL TRANSFORMATION"
_ICLR.cc/2020/Conference — Reject_

### Official Review · AnonReviewer1 · 2019-10-23
**Official Blind Review #1**

**Rating:** 6

**Review:**

This paper proposes a spectral graph neural network based on a graph kernel to predict graph evolution. The overall idea is interesting and the biggest advantage is scalability of the framework to large graphs in terms of time and parameter complexity.
The major drawback of the paper is the lack of experimentation with real datasets. Based on the results from four datasets they used, the efficacy of their proposed method is unclear. The synthetic datasets are hard to admit in this case.
Note: I could not verify the theory in detail yet.

**Experience Assessment:**

I have read many papers in this area.

**Review Assessment: Checking Correctness Of Derivations And Theory:**

I assessed the sensibility of the derivations and theory.

**Review Assessment: Checking Correctness Of Experiments:**

I carefully checked the experiments.

**Review Assessment: Thoroughness In Paper Reading:**

I read the paper at least twice and used my best judgement in assessing the paper.

---

> ### Author Response · Authors · 2019-11-07
> **Response to the comments on our experiments**
>
> Thanks a lot for the reviewer's comments. We devoted extensive efforts to the experiments and hence deeply believe that our experiments on 15 datasets (4 real-world datasets + 11 synthetic datasets) against 7 comparison methods are sufficient to demonstrate the effectiveness and efficiency of our methods.
>
> More details are as follows:
>
> First, the experiments on four real-world datasets demonstrate that the proposed model outperforms all the comparison methods in accuracy and efficiency significantly. Specifically, Table 3 shows that our method achieved the best performance on 3 out of 4 real-world datasets when being compared with 7 other state-of-the-art methods, and achieved the second-best in the remaining one dataset. Our method also obtained the best overall performance. Moreover, our method is 40 to 1000 times faster than the most competitive comparison methods as shown in Table 4. As also mentioned by the reviewer "the biggest advantage of our method is scalability in terms of time and complexity", we believe our above real-world experiments are sufficient to support it.
>
> Second, the synthetic dataset also showed that our method achieved the highest performance in all the datasets. As explained in the caption of Table 2 and in Section 5.2.1, the results marked as "GS" are the performance achieved by the "gold standard" (i.e., the prediction model is the data generator itself). So the purpose of the experiment is to validate how close our method's performance is to the gold standard's. And it can be seen that our method achieved the best performance (i.e., closest to gold standard's) in 8 out of 11 synthetic datasets. And our overall performance is 50% higher than the best performer (i.e., C-DGT) among the comparison methods. This strongly demonstrates that our end-to-end methods are effective in various datasets even with different types of transformations.

---

### Official Review · AnonReviewer2 · 2019-10-25
**Official Blind Review #2**

**Rating:** 3

**Review:**

The paper proposed a method to model the graph topological evolution from the spectral domain by developing a new generalized graph kernel. The new graph kernels cover many existing graph kernels as well as their combination and composition as special cases. The idea of spectral graph translation and its integration with deep learning is interesting, especially considering that most previous spectral graph neural networks only transform the graph signal instead of graph structures. However, I do have some concerns of papers.

1.	My major concern is the soundness of keeping eigenvectors unchanged in the evolution. Although the authors claim that in previous studies eigenvectors are found stable in evolution, it is very counter-intuitive, and I am not sure it is the case for all types of graphs. Let us look at the proof of Lemma 3.1, obviously $L^\prime$ does not necessarily have the same eigenvectors, and $U^TL^\prime U$ is not a diagonal matrix, so this loss is actually very large in many cases. That is to say, the evolution model does not have enough expressive power to recover the $L^\prime$.
2.	Spectral graph translation looks interesting, but the main idea comes from Kunegis et al. (2010). Despite of a new designed graph kernel and adding nonlinear activations, the contributions seem not so significant.
3.	In Kunegis et al. (2010), they consider the evolution of adjacency matrix $A$, but in this paper the authors use the Laplacian matrix $L$. If there any reason to make this choice? Also, I think some of the conclusions (e.g. stable eigenvectors) in Kunegis et al. (2010) may not work since $L$ is used instesd of $A$.
4.    The correlation metric is acceptable, but it will be much better if the authors can do more analysis. For example, why not add the link prediction task as in Kunegis et al. (2010)? BTW, is correlation analysis used before in previous graph evolution papers? (if so, please add a reference)


**Experience Assessment:**

I have published one or two papers in this area.

**Review Assessment: Checking Correctness Of Derivations And Theory:**

I assessed the sensibility of the derivations and theory.

**Review Assessment: Checking Correctness Of Experiments:**

I assessed the sensibility of the experiments.

**Review Assessment: Thoroughness In Paper Reading:**

I read the paper at least twice and used my best judgement in assessing the paper.

---

> ### Author Response · Authors · 2019-11-07
> **Authors' Response to Reviewer #2**
>
> We appreciate the comments from the reviewer very much.
>
> 1. For the first concern. Actually our paper is neither targeted at nor has claimed to handle all types of graph evolution process. Instead, we only aim at those situations whose eigenvectors do not change or do not change much. And it can be seen in our Lemma 3.1, our method has good expressiveness for such situations, with significant contributions in the following aspects:
>
> i. Such situations, where eigenvectors do not change much during graph evolution, widely exist in the real world. First, there are hundreds of commonly-used graph kernels (e.g., diffusion kernels) who are aiming to explain those phenomena whose eigenvectors keep unchanged during evolution, as exemplified in Table 1. Beyond that, our method is an end-to-end framework that achieves higher expressiveness over all of them. Second, in many (and ever-increasing) domains, people observed that the eigenvectors do not change much. For example, in the “befriending process” in a social network and in structural to functional connectivity transformation in the neuroscience domain. Moreover, our experimental results in 15 datasets with different types of graph process demonstrates the effectiveness of the proposed method for various types of graph evolution process.
>
> ii. Our method enjoys huge efficiency advantages for the situations when eigenvectors do not change. Namely, the complexity is reduced to linear to the graph size, compared with the (at least) quadratic complexity of traditional methods.
>
> iii. It is convenient to verify if a specific application is suitable for our method because it is easy to see whether a graph evolution has the eigenvectors unchanged or not. More importantly, our method can help to identify if an evolution process has an unchanged eigenvector or not. Specifically, when our methods achieve high prediction performance, then the process tends to be spectral evolution process with no (or little) change in eigenvectors. This is because as shown in Lemma 3.1, if our model, which enforces low error in Equation (1), also leads to low error in $\|F(U\Lambda U^T) – L’\|^2$, then U^T L’ U is close to a diagonal matrix.
>
> 2. For the second concern, we believe our contribution is significant. This paper is much more than proposing merely a new kernel, but instead propose an end-to-end framework that can fit any existing and (unknown) kernels as well as their compositions and summations. This is radically different because the previous graph kernels are still prescribed models that require human labors to tailor a predefined kernel to targeted data by prior knowledge or heuristics. But our model is the first which can purely rely on the data and demonstrate our dominating expressiveness over traditional kernels by extensive experiments on 15 datasets and by theoretical discussions in Lemma 3.2. Also, we believe that being as the first work for deep spectral graph translation, it opens a new window for the deep graph learning community.
>
> 3. For the third concern, we want to clarify that our method is generic to whichever it is $L$ or $A$ (this is why we mentioned “without loss of generalizability” in Lemma 3.1, but we will explicitly mention this in the revision. This means we can also use $A$ instead of $L$. The reason why we prefer $L$ a little bit in this paper is because in deep graph learning domain, graph Laplacian $L$ seems more commonly used and its eigen-decomposition has higher popularity.
>
> 4. For the last concern of the reviewer, we will have no problem to add more analysis using more metrics, such as RMSE. We are working on that and try our best to give the updates by the end of the rebuttal session. The reason why we did not use link prediction is due to the nature of our real-world applications. Specifically, for our brain network application, researchers in the domain all achieve the prediction of functional connectivity all at once. Similarly, the other application on the authentication networks also focuses on whole graph generation. Following the domains’ inherent setting will not only ensure that our experiment is practically meaningful but also ensure we are able to compare with the state-of-the-art methods in their domain.  Correlation is commonly used for graph evolution papers, such as:
>
> Abdelnour, Farras, et al. "Functional brain connectivity is predictable from anatomic network's Laplacian eigen-structure." Neuroimage 172 (2018): 728-739.
>
> Honey, C. J., et al. "Predicting human resting-state functional connectivity from structural connectivity." Proceedings of the National Academy of Sciences (PNAS) 106.6 (2009): 2035-2040.
>
> Xiaojie Guo, Liang Zhao, Cameron Nowzari, Setareh Rafatirad, Houman Homayoun, and Sai Dinakarrao. Deep Multi-attributed Graph Translation with Node-Edge Co-evolution. The 19th International Conference on Data Mining (ICDM 2019), Beijing, China.

---

> > ### Author Response · Authors · 2019-11-11
> > **Addition of more analysis with additional evaluation metric as suggested by Reviewer #2**
> >
> > Dear Reviewer #2, following Item 4 in your comments, we have added more analysis on the performance evaluation of our method and comparison methods on all four real-world datasets. This new evaluation is based on the R2 score, which is a widely-used metric for prediction performance evaluation. Please see them in Table 5 in the Appendix, along with the discussions in Appendix A3.
> >
> > Briefly speaking, as shown in Table 5, our method GSEN achieves the best performance in 3 out of 4 datasets and is still highly competitive in the remaining one dataset. Our GSEN also achieves the best performance on average with a large margin when compared with other methods.

---

### Decision · Program_Chairs · 2019-12-19

**Decision:**

Reject

**Comment:**

The reviewers kept their scores after the author response period, pointing to continued concerns with methodology, needing increased exposition in parts, and not being able to verify theoretical results. As such, my recommendation is to improve the clarity around the methodological and theoretical contributions in a revision.